# Genetically predicted high IGF-1 levels showed protective effects on COVID-19 susceptibility and hospitalization: a Mendelian randomisation study with data from 60 studies across 25 countries

Xinxuan Li[1], Yajing Zhou[1], Shuai Yuan[1,2], Xuan Zhou[1], Lijuan Wang[1], Jing Sun[1], Lili Yu[1], Jinghan Zhu[3], Han Zhang[1], Nan Yang[1], Shuhui Dai[1], Peige Song[4], Susanna C Larsson[2,5], Evropi Theodoratou[6,7†], Yimin Zhu[1†], Xue Li[1*†]

[1]Department of Big Data in Health Science School of Public Health, Center of Clinical Big Data and Analytics of The Second Affiliated Hospital, Zhejiang University School of Medicine, Hangzhou, China; [2]Unit of Cardiovascular and Nutritional Epidemiology, Institute of Environmental Medicine, Karolinska Institutet, Stockholm, Sweden; [3]The Second School of Clinical Medicine, Southern Medical University, Guangzhou, China; [4]School of Public Health and Women's Hospital, Zhejiang University School of Medicine, Hangzhou, China; [5]Unit of Medical Epidemiology, Department of Surgical Sciences, Uppsala University, Uppsala, Sweden; [6]Centre for Global Health, Usher Institute, University of Edinburgh, Edinburgh, United Kingdom; [7]Cancer Research UK Edinburgh Centre, Medical Research Council Institute of Genetics and Cancer, University of Edinburgh, Edinburgh, United Kingdom

*For correspondence: xue.li@ed.ac.uk

†These authors contributed equally to this work

## Abstract

**Background:** Epidemiological studies observed gender differences in COVID-19 outcomes, however, whether sex hormone plays a causal in COVID-19 risk remains unclear. This study aimed to examine associations of sex hormone, sex hormones-binding globulin (SHBG), insulin-like growth factor-1 (IGF-1), and COVID-19 risk.

**Methods:** Two-sample Mendelian randomization (TSMR) study was performed to explore the causal associations between testosterone, estrogen, SHBG, IGF-1, and the risk of COVID-19 (susceptibility, hospitalization, and severity) using genome-wide association study (GWAS) summary level data from the COVID-19 Host Genetics Initiative (N=1,348,701). Random-effects inverse variance weighted (IVW) MR approach was used as the primary MR method and the weighted median, MR-Egger, and MR Pleiotropy RESidual Sum and Outlier (MR-PRESSO) test were conducted as sensitivity analyses.

**Results:** Higher genetically predicted IGF-1 levels have nominally significant association with reduced risk of COVID-19 susceptibility and hospitalization. For one standard deviation increase in genetically predicted IGF-1 levels, the odds ratio was 0.77 (95% confidence interval [CI], 0.61–0.97, p=0.027) for COVID-19 susceptibility, 0.62 (95% CI: 0.25–0.51, p=0.018) for COVID-19 hospitalization, and 0.85 (95% CI: 0.52–1.38, p=0.513) for COVID-19 severity. There was no evidence that testosterone, estrogen, and SHBG are associated with the risk of COVID-19 susceptibility, hospitalization, and severity in either overall or sex-stratified TSMR analysis.

**Conclusions:** Our study indicated that genetically predicted high IGF-1 levels were associated with decrease the risk of COVID-19 susceptibility and hospitalization, but these associations did not survive the Bonferroni correction of multiple testing. Further studies are needed to validate the

findings and explore whether IGF-1 could be a potential intervention target to reduce COVID-19 risk.

**Funding:** We acknowledge support from NSFC (LR22H260001), CRUK (C31250/A22804), SHLF (Hjärt-Lungfonden, 20210351), VR (Vetenskapsrådet, 2019-00977), and SCI (Cancerfonden).

## Editor's evaluation

Using publicly available genetic data, Li and colleagues tested the association and inferred the causality of genetic variants predicted to alter the levels of testosterone, estrogen, SHBG, or IGF-1, against susceptibility, severity and outcome of SARS-Cov2 infection. The main strength of the study is the large cohort which adds to the robustness of the data.

## Introduction

The COVID-19 pandemic has emerged as the most important health concern across the globe since December 2019. A notable finding that has been noted in many affected countries is a male predominance of COVID-19-related hospitalization and death (*Grasselli et al., 2020*; *Peckham et al., 2020*). Globally, more than 60% of deaths from COVID-19 are reported in males (*Richardson et al., 2020*). This epidemiological pattern indicates the need for urgent public health actions, as well as for further investigations on the contributing factors of sex differences in COVID-19 risk and its underlying biological mechanisms.

Sex hormones play important roles in the immune response in which estrogen was thought to be immune boosting and testosterone to be immunosuppressing (*Strope et al., 2020*). Due to the higher levels of testosterone in male than female, it has been hypothesized that testosterone might be a promoter of SARS-CoV-2 infection and progression in males, considering the regulatory effect of androgen receptor (AR) and testosterone on the transcription of a transmembrane protease serine 2, which is a critical factor enabling cellular infection by coronaviruses, including SARS-CoV-2 (*Peckham et al., 2020*; *Pozzilli and Lenzi, 2020*; *Cattrini et al., 2020*). Estrogen has been shown not only to enhance immunological markers and response, but also to be linked to T-cell proliferation, which might be involved in the immune response to the infection of SARS-CoV-2 (*Taneja, 2018*). Most hormone (about 60%) is tightly bound to sex hormone-binding globulin (SHBG), which is an important regulator of the bioactivities of estrogens and testosterone (*Raverot et al., 2010*; *Dimou et al., 2021*). In addition, sex hormone signaling could also regulate the insulin-like growth factor (IGF-1) concentrations, which were also reported to be associated with acute respiratory distress syndrome (*Ahasic et al., 2012*). It is therefore hypothesized that sex hormone and its related biomarkers might contribute to the sex difference of COVID-19 outcomes. A number of observational studies examined the associations between sex hormones and COVID-19 risk, however, the causality of these associations remains unestablished due to potential limitations of observational studies (e.g., residual confounding and reverse causality) and lack of high-quality data from randomized trials (*Tsang et al., 2016*).

Mendelian randomization (MR) analysis is an epidemiological approach that can strengthen the casual inference by utilizing genetic variants as instrumental variables to mimic biological effects of related biomarkers (*Burgess and Thompson, 2015*). Here, we conducted a two-sample MR (TSMR) study to explore the causal associations testosterone, estrogen, SHBG, and IGF-1 with the risk of COVID-19 (susceptibility, hospitalization, and severity) using genome-wide association study (GWAS) summary level data from the COVID-19 Host Genetics Initiative (COVID-19 HGI). Sex-stratified MR analyses for testosterone and estradiol were further performed to explore the associations in males and females separately.

## Materials and methods

### Study design

We firstly conducted a TSMR analysis to explore the causal links between testosterone, estrogen, SHBG, IGF-1, and the risk of COVID-19 (susceptibility, hospitalization, and severity), based on GWAS summary level data from COVID-19 HGI. We then performed sex-stratified MR analysis to

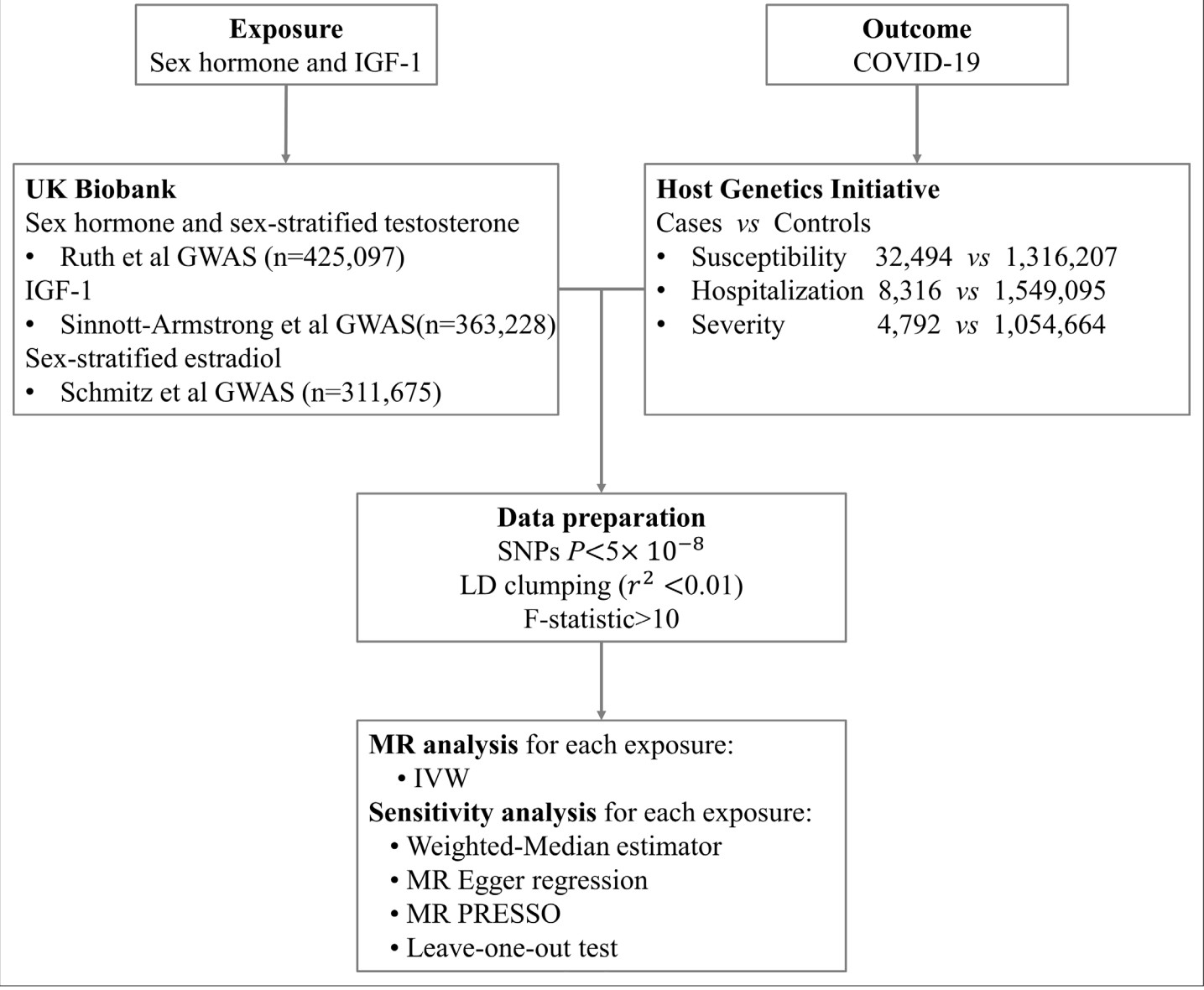

**Figure 1.** Overall study design. Abbreviation: IGF-1, insulin-like growth factor-1; GWAS, genome-wide association study; SNP, single-nucleotide polymorphism; LD, linkage disequilibrium; IVW, inverse variance weighting; MR, Mendelian randomization.

further examine the associations between genetically determined circulating levels of testosterone and estrogen and COVID-19 outcomes in males and females separately. The design of this study is explained in *Figure 1*.

## Genetic instruments of testosterone, estradiol, SHBG, and IGF-1

Single-nucleotide polymorphisms (SNPs) associated with testosterone, estradiol, SHBG, and IGF-1 levels were identified from genome-wide association analyses in up to 425,097 participants of European ancestry (*Ruth et al., 2020*; *Sinnott-Armstrong et al., 2021*). Sex-stratified SNPs related to estradiol were obtained from a GWAS including 147,690 males and 163,985 females in UK Biobank (*Schmitz et al., 2021*). We restricted the analysis to SNPs in linkage equilibrium which were identified in the relevant GWAS at $p<5 \times 10^{-8}$ clumped on $r^2=0.01$ within 10,000 kb using the 1000 genomes reference panel (*Hemani et al., 2018*) to ensure sufficient statistical effectiveness. Among those pairs of SNPs that had LD $r^2$ above the specified threshold ($r^2=0.01$), only the SNP with the lower p value would be retained. SNPs absent from the LD reference panel were also removed. To test whether

**Table 1.** Sources of data for Mendelian randomization analysis in COVID-19 HGI.

| Phenotype | Participants | | |
|---|---|---|---|
| | Meta-analysis of 35 GWAS performed in individuals of European ancestry | | |
| | **Cases**: 32,494 individuals with COVID-19 by laboratory confirmation, chart review, or self-report | | |
| Susceptibility | **Controls**: 1,316,207 individuals without confirmation or history of COVID-19 | | |
| | Meta-analysis of 23 GWAS performed in individuals of European ancestry | | |
| | **Cases**: 8316 hospitalized individuals with COVID-19 | | |
| Hospitalization | **Controls**: 1,549,095 individuals without confirmation or history of COVID-19 | | |
| | Meta-analysis of 14 GWAS performed in individuals of European ancestry | | |
| | **Cases**: 4792 SARS-CoV-2 infected hospitalized individuals who died or required respiratory support (intubation, CPAP, BiPAP, continuous external negative pressure, high flow nasal cannula). | | |
| Severity | **Controls**:1,054,664 individuals without confirmation or history of COVID-19 | | |

Notes: COVID-19 outcomes are taken from the COVID-19 HGI.
HGI = Host Genetics Initiative. GWAS = genome-wide association study. UKB = UK Biobank. CPAP = continuous positive airway pressure ventilation. BiPAP = bilevel positive airway pressure ventilation.

there was a weak instrumental variable bias, namely genetic variants selected as instrumental variables had a weak association with exposure, we calculated the F statistic if it is much greater than 10 for the instrument-exposure association, the possibility of weak instrumental variable bias is small. These analyses were conducted using the R package 'TwoSampleMR' (*Yavorska and Burgess, 2017*). Consequently, a total of 320, 316, 7, and 18 SNPs were used as instrumental variables for SHBG, testosterone, estradiol, and IGF-1, respectively. Given that genetic variants predicting testosterone and estradiol levels differ for men and women, we selected sex-specific SNPs for testosterone (130 SNPs in males, 151 SNPs in females) and estradiol (10 SNPs in males and females) separately for MR sensitivity analyses. Detailed information on the genetic instruments were provided in *Supplementary file 1a-d*. We used the STROBE case-control checklist when writing our report (*von Elm et al., 2014*).

## Data source from COVID-19 HGI

We obtained the summary level data of COVID-19 susceptibility, hospitalization, and severity from the COVID-19-HGI GWAS meta-analyses of data across 60 studies from 25 countries (Round 5, European population) where UK Biobank data were excluded (*COVID-19 Host Genetics Initiative, 2020*). The HGI dataset included 1,348,701 participants (32,494 laboratory-confirmed cases of SARS-CoV-2 infection and 1,316,207 population controls) for COVID-19 susceptibility, 1,557,411 participants (8316 hospitalized COVID-19 patients and 1,549,095 population controls) for COVID-19 hospitalization, and 1,059,456 participants (4792 very severe respiratory-confirmed COVID-19 cases and 1,054,664 controls) for COVID-19 severity. COVID-19-HGI defined very severe respiratory-confirmed COVID-19 cases as patients hospitalized for laboratory-confirmed SARS-CoV-2 infection who died or were given respiratory support. The characteristics of the participants are shown in *Table 1*.

## TSMR analyses

We applied the inverse variance weighted (IVW) method under the random-effects model as the primary MR analysis. We performed sensitivity analyses, including the weighted median, MR-Egger regression, leave-one-out analysis, and MR Pleiotropy RESidual Sum and Outlier (MR-PRESSO) methods, to examine the consistency of associations and to detect and correct for potential pleiotropy. The weighted median method was performed to provide unbiased causal estimates if at least 50% instrumental variables were valid (*Bowden et al., 2016*). MR-Egger regression was used to observe and correct potential directional pleiotropy, which was assessed by its intercept test (*Bowden et al., 2015*). MR-PRESSO method can detect SNP outliers and estimate the association after removal of these outliers. The differences in estimates between before and after outlier removal were examined by the embedded distortion test (*Wu et al., 2020*). Cochrane's Q value was used to assess the

heterogeneity among estimates of genetic instruments and the p value for intercept in MR-Egger was used to detect horizontal pleiotropy (*Bowden et al., 2015*). All statistical analyses were two-sided and performed in R 4.0.4 software using the R package TwoSampleMR and MR-PRESSO (*Yavorska and Burgess, 2017*).

### Sensitivity analyses

We additionally used the SNP rs7173595 in *CYP19A1* gene, which encodes aromatase, an enzyme that converts androgens to estrogens. Rs7173595 has previously been shown to be strongly associated with serum E2 levels in GWAS of men (*Ruth et al., 2020*; *Eriksson et al., 2018*) and postmenopausal women (*Thompson et al., 2016*). This SNP was also associated with serum E2 in 25,502 premenopausal European women (<50 years of age and not reporting a hysterectomy or that menopause has occurred) in UK Biobank. The associations of serum E2 instrumented by rs7173595 in the *CYP19A1* gene region with COVID-19 outcomes were estimated using the Wald ratio method. We further performed a sensitivity analysis using a list of genetic instruments consisting of 10 correlated SNPs ($r^2 < 0.4$) located in the *IGF-1* gene region (genomic position on build GRCh37/hg19: chromosome 12:102789652–102874341) and associated with IGF-1 levels at the genome-wide significance level. A matrix of linkage disequilibrium among these SNPs was introduced in the MR analysis model. To control potential data confounder, we selected SNPs associated with testosterone, estrogen, SHBG, and IGF-1 only, excluding SNPs associated with BMI which is thought to be a causal risk factor for COVID-19 (*Freuer et al., 2021*) at the threshold of $5\times10^{-8}$ in European ancestry samples by querying PhenoScanner (*Yavorska and Burgess, 2017*). SNPs in estrogen were not excluded because their irrelevance to BMI.

### Results

*Table 2* presents the TSMR estimates for the associations between sex hormones, SHBG, IGF-1, and the risk of COVID-19 susceptibility, hospitalization, and severity based on the data from HGI. Higher genetically predicted IGF-1 levels have nominally significant association with reduced risk of COVID-19 susceptibility and hospitalization. For one standard deviation increase in genetically predicted IGF-1 levels, the odds ratio was 0.77 (95% confidence interval [CI], 0.61–0.97, p=0.027) for COVID-19 susceptibility, 0.62 (95% CI: 0.25–0.51, p=0.018) for COVID-19 hospitalization, and 0.85 (95% CI: 0.52–1.38, p=0.513) for COVID-19 severity. Associations of IGF-1 levels with COVID-19 susceptibility and hospitalization were not statistically significant after Bonferroni correction, albeit showing a nominal significance at p<0.05. No outlying SNPs were identified by MR-PRESSO analyses. Estimates from the MR-Egger and weighted mode analyses were in the same direction as those from the IVW analysis (*Figure 2*, *Figure 2—figure supplement 1*, *Figure 2—figure supplement 2*). The MR-Egger intercept p was 0.614 and 0.595 for susceptibility and hospitalization, respectively, indicating the absence of directional pleiotropy. The associations remained directionally consistent in the sensitivity analysis based on SNPs located in the *IGF-1* gene region as instrumental variables with risk of COVID-19 susceptibility (OR = 0.99, 95% CI: 0.91–1.07, p=0.777), hospitalization (OR = 0.90; 95% CI: 0.74–1.10, p=0.645), and severity (OR = 1.01; 95% CI: 0.82–1.24, p=0.415) (*Table 3*).

In the analyses based on data from the genetic consortia, we found no causal associations of genetically predicted testosterone with the risk of COVID-19 susceptibility (OR = 0.94; 95% CI: 0.83–1.06, p=0.309), hospitalization (OR = 0.82; 95% CI: 0.64–1.04, p=0.103), risk of severity (OR = 0.83; 95% CI: 0.60–1.15, p=0.256). Null association was also noticed between SHBG and COVID-19 susceptibility (OR = 0.91; 95% CI: 0.80–1.04, p=0.182), hospitalization (OR = 0.86; 95% CI: 0.66–1.11, p=0.255), risk of severity (OR = 0.92; 95% CI: 0.65–1.29, p=0.618). Overall, no significant associations between testosterone, estrogen, SHBG, and COVID-19 outcomes were observed from TSMR analyses. Sex-specific associations of genetically testosterone and estradiol levels with COVID-19 risk (*Table 4*) were still nonsignificant. We noticed that the p for intercept in MR-Egger regression analysis was more than 0.05 for both genders, and no outlier was detected. Genetic predisposition to higher serum E2 levels proxied by rs7173595 in the *CYP19A1* gene was not associated with the risk of COVID-19 susceptibility (OR = 0.32; 95% CI, 0.06–1.80, p = 0.195), hospitalization (OR = 0.28; 95% CI: 0.01–6.46, p=0.426), and severity (OR = 0.22; 95% CI: 0.00–12.73, p=0.469) in females; similarly, the associations remained directionally consistent in males with susceptibility (OR = 0.37; 95% CI, 0.08–1.67,

**Table 2.** Sex hormones, SHBG, IGF-1, and COVID-19 outcomes in Mendelian randomization (MR) analyses.

| Exposure | Method | Susceptibility | | | | | Hospitalization | | | | | Severity | | | | |
|---|---|---|---|---|---|---|---|---|---|---|---|---|---|---|---|---|
| | | SNPs | OR (95% CI) | p Effect | p Heterogeneity | p Intercept | SNPs | OR (95% CI) | p Effect | p Heterogeneity | p Intercept | SNPs | OR (95% CI) | p Effect | p Heterogeneity | p Intercept |
| Testosterone | IVW | | 0.94 (0.83, 1.06) | 0.309 | 0.006 | – | | 0.82 (0.64, 1.04) | 0.103 | 0.055 | – | | 0.83 (0.60, 1.15) | 0.256 | 0.041 | – |
| | MR-Egger | | 0.93 (0.76, 1.12) | 0.430 | 0.005 | 0.860 | | 0.79 (0.55, 1.15) | 0.217 | 0.051 | 0.819 | | 0.78 (0.48, 1.27) | 0.313 | 0.038 | 0.732 |
| | Weighted median | | 0.89 (0.71, 1.12) | 0.329 | – | – | | 0.81 (0.52, 1.28) | 0.370 | – | – | | 0.71 (0.40, 1.26) | 0.246 | – | – |
| | Simple mode | | 1.13 (0.73, 1.77) | 0.584 | – | – | | 0.77 (0.27, 2.20) | 0.623 | – | – | | 0.44 (0.09, 2.18) | 0.316 | – | – |
| | Weighted mode | | 0.91 (0.77, 1.08) | 0.300 | – | – | | 0.77 (0.52, 1.13) | 0.180 | – | – | | 0.65 (0.40, 1.05) | 0.081 | – | – |
| | MR-PRESSO | 315 | 0.94 (1.06, 0.84) | – | – | – | 303 | 0.82 (1.04, 0.65) | – | – | – | 316 | 0.83 (1.15, 0.59) | – | – | – |
| SHBG | IVW | | 0.91 (0.80, 1.04) | 0.182 | 0.002 | – | | 0.86 (0.66, 1.11) | 0.255 | 0.087 | – | | 0.92 (0.65, 1.29) | 0.618 | 0.096 | – |
| | MR-Egger | | 0.96 (0.78, 1.18) | 0.708 | 0.002 | 0.494 | | 0.83 (0.57, 1.22) | 0.352 | 0.081 | 0.818 | | 0.92 (0.56, 1.51) | 0.730 | 0.090 | 0.994 |
| | Weighted median | | 0.90 (0.72, 1.13) | 0.360 | – | – | | 0.82 (0.52, 1.29) | 0.391 | – | – | | 0.72 (0.41, 1.27) | 0.255 | – | – |
| | Simple mode | | 1.09 (0.66, 1.81) | 0.735 | – | – | | 1.18 (0.40, 3.44) | 0.767 | – | – | | 1.16 (0.25, 5.41) | 0.850 | – | – |
| | Weighted mode | | 0.94 (0.78, 1.14) | 0.547 | – | – | | 0.81 (0.56, 1.18) | 0.279 | – | – | | 0.79 (0.47, 1.33) | 0.376 | – | – |
| | MR-PRESSO | 319 | 0.91 (1.05, 0.80) | – | – | – | 309 | 0.86 (1.11, 0.67) | – | – | – | 320 | 0.91 (1.28, 0.65) | – | – | – |
| Estradiol | IVW | 7 | 0.54 (0.15, 1.94) | 0.346 | 0.188 | – | 7 | 0.87 (0.11, 6.70) | 0.895 | 0.769 | – | 7 | 0.50 (0.03, 7.64) | 0.620 | 0.987 | – |
| | MR-Egger | | 0.73 (0.04, 14.11) | 0.845 | 0.123 | 0.830 | | 0.34 (0.00, 29.54) | 0.657 | 0.685 | 0.662 | | 0.04 (0.00, 17.04) | 0.345 | 1.000 | 0.401 |
| | Weighted median | | 0.36 (0.10, 1.35) | 0.130 | – | – | | 0.35 (0.03, 4.21) | 0.407 | – | – | | 0.30 (0.01, 7.26) | 0.458 | – | – |
| | Simple mode | | 0.29 (0.03, 2.60) | 0.313 | – | – | | 0.71 (0.01, 44.94) | 0.875 | – | – | | 0.33 (0.00, 43.56) | 0.673 | – | – |
| | Weighted mode | | 0.34 (0.07, 1.73) | 0.241 | – | – | | 0.38 (0.03, 4.81) | 0.482 | – | – | | 0.29 (0.01, 9.43) | 0.511 | – | – |
| | MR-PRESSO | | 0.54 (1.94, 0.15) | – | – | – | | 0.87 (3.93, 0.19) | – | – | – | | 0.51 (1.52, 0.17) | – | – | – |

*Table 2 continued on next page*

Table 2 continued

| Exposure | Method | Susceptibility | | | | | Hospitalization | | | | | Severity | | | | |
|---|---|---|---|---|---|---|---|---|---|---|---|---|---|---|---|---|
| | | SNPs | OR (95% CI) | p Effect | p Heterogeneity | p Intercept | SNPs | OR (95% CI) | p Effect | p Heterogeneity | p Intercept | SNPs | OR (95% CI) | p Effect | p Heterogeneity | p Intercept |
| | IVW | | 0.77 (0.61, 0.97) | 0.027 | 0.175 | – | | 0.62 (0.25, 0.51) | 0.018 | 0.715 | – | | 0.85 (0.52, 1.38) | 0.513 | 0.601 | – |
| | MR-Egger | | 0.84 (0.56, 1.26) | 0.408 | 0.145 | 0.614 | | 0.72 (0.37, 1.38) | 0.336 | 0.668 | 0.595 | | 1.45 (0.67, 3.10) | 0.358 | 0.758 | 0.096 |
| | Weighted median | | 0.76 (0.57, 1.02) | 0.071 | – | – | | 0.75 (0.44, 1.28) | 0.294 | – | – | | 0.76 (0.38, 1.53) | 0.446 | – | – |
| | Simple mode | | 0.64 (0.39, 1.05) | 0.097 | – | – | | 0.66 (0.30, 1.45) | 0.318 | – | – | | 0.82 (0.27, 2.47) | 0.730 | – | – |
| | Weighted mode | | 0.77 (0.58, 1.02) | 0.084 | – | – | | 0.71 (0.44, 1.17) | 0.199 | – | – | | 0.70 (0.35, 1.38) | 0.319 | – | – |
| IGF-1 | MR-PRESSO | 16 | 0.77 (0.98, 0.61) | – | – | – | 16 | 0.62 (0.88, 0.43) | – | – | – | 18 | 0.85 (1.34, 0.54) | – | – | – |

SNP = single-nucleotide polymorphism. OR = odds ratio. CI = confidence interval. IVW = inverse variance weighting. SHBG = sex hormones-binding globulin. IGF-1 = insulin-like growth factor-1.

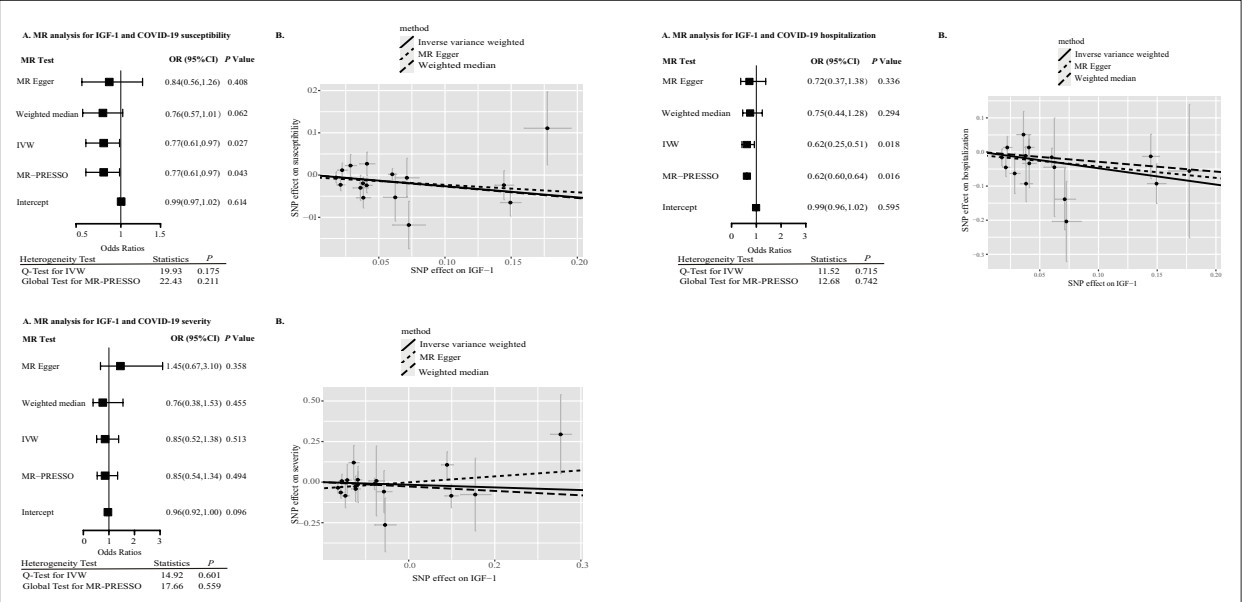

**Figure 2.** IGF-1 and COVID-19 outcomes in Mendelian randomization (MR) analyses. Abbreviation: IGF-1, insulin-like growth factor-1; SNP, single-nucleotide polymorphism; IVW, inverse variance weighting; OR, odds ratio; CI, confidence interval.

The online version of this article includes the following figure supplement(s) for figure 2:

**Figure supplement 1.** Leave-one-out plot for IGF-1 and COVID-19 susceptibility, hospitalization and severity in Mendelian randomization analysis.

**Figure supplement 2.** Funnel plot for IGF-1 and COVID-19 susceptibility, hospitalization and severity in Mendelian randomization analysis.

p = 0.195), hospitalization (OR = 0.33; 95% CI: 0.02–5.11, p=0.426), and severity (OR = 0.27; 95% CI: 0.01–9.26, p=0.469) (*Table 5*). As shown in *Table 6*, after removing SNPs associated with BMI, we found similar associations of genetically predicted IGF-1 levels with the risk of COVID-19 susceptibility (OR = 0.76; 95% CI: 0.60–0.96, p=0.021), hospitalization (OR = 0.61; 95% CI: 0.41–0.90, p=0.014), risk of severity (OR = 0.84; 95% CI: 0.52–1.38, p=0.497) in which we detected no moderate heterogeneity, and no indication of horizontal pleiotropy in MR-Egger, and no outlier in MR-PRESSO analyses. No causal associations of genetically predicted testosterone and SHBG with COVID-19 were found, but the directions were consistent with results in *Table 2*.

## Discussion

In this study, we assessed whether there were any causal associations between sex hormone-related biomarkers and the risk of COVID-19 outcomes. We found suggestive evidence for associations between genetic liability to high IGF-1 levels and decreased risk of COVID-19 susceptibility and hospitalization. Our findings suggest a potential role of IGF-1 in COVID-19 risk and have implications for tailored treatment of COVID-19 patients.

Our MR findings were consistent with the multiple epidemiological studies that reported a nominal association between measured IGF-1 levels and COVID-19 illness. There is one observational study that demonstrated an inverse association between pre-diagnostic circulating levels of IGF-1 and COVID-19 mortality risk among COVID-19 patients in UK Biobank (*Fan et al., 2021*). Another observational study in Greece reported lower IGF-1 levels in critically ill COVID-19 patients compared to their counterparts with less severe disease or without COVID-19 (*Ilias et al., 2021*). A single-cell analysis revealed that the exhaustion of CD8[+] T cells together with several cytokines including IGF-1 was associated with the pathogenesis of severe SARS-CoV-2 infection (*He et al., 2021*). Our MR analyses found a negative association between genetically determined high circulating IGF-1 levels and decreased risk of COVID-19 susceptibility and hospitalization, indicating IGF-1 may be a protective factor of COVID-19 risk.

IGF-1 has been found to be pro-survival/anti-aging, anti-inflammatory, and antioxidant with neuro- and hepatoprotective properties. A study by the Narasaraju group demonstrated that IGF-1 plays

**Table 3.** Sensitive analysis between serum IGF-1 levels instrumented by 10 SNPs in the IGF-1 gene region and COVID-19 outcomes.

| Method | Susceptibility | | | | Hospitalization | | | | Severity | | | |
|---|---|---|---|---|---|---|---|---|---|---|---|---|
| | OR (95% CI) | p Effect | p Heterogeneity | p Intercept | OR (95% CI) | p Effect | p Heterogeneity | p Intercept | OR (95% CI) | p Effect | p Heterogeneity | p Intercept |
| IVW | 0.99 (0.91, 1.07) | 0.777 | 0.596 | – | 0.90 (0.74, 1.10) | 0.645 | 0.104 | – | 1.01 (0.82, 1.24) | 0.415 | 0.437 | – |
| MR-Egger | 0.99 (0.93, 1.05) | 0.732 | 0.541 | 0.527 | 0.97 (0.84, 1.11) | 0.338 | 0.108 | 0.375 | 1.09 (0.92, 1.30) | 0.953 | 0.372 | 0.590 |
| Weighted median | 1.01 (0.96, 1.06) | 0.739 | – | – | 0.97 (0.86, 1.10) | 0.620 | – | – | 1.05 (0.93, 1.20) | 0.310 | – | – |
| Simple mode | 0.98 (0.89, 1.08) | 0.685 | – | – | 1.12 (0.88, 1.43) | 0.395 | – | – | 1.16 (0.88, 1.51) | 0.316 | – | – |
| Weighted mode | 0.98 (0.92, 1.05) | 0.596 | – | – | 0.94 (0.82, 1.09) | 0.439 | – | – | 1.12 (0.92, 1.37) | 0.279 | – | – |

IGF-1 = insulin-like growth factor-1. SNP = single-nucleotide polymorphism. IWW = inverse variance weighting. OR = odds ratio. CI = confidence interval.

**Table 4.** Sex-specific associations of genetically testosterone and estradiol levels with COVID-19 risk.

| Exposure | Method | Susceptibility | | | | Hospitalization | | | | Severity | | | |
|---|---|---|---|---|---|---|---|---|---|---|---|---|---|
| | | Male | | Female | | Male | | Female | | Male | | Female | |
| | | OR (95% CI) | p | OR (95% CI) | p | OR (95% CI) | p | OR (95% CI) | p | OR (95% CI) | p | OR (95% CI) | p |
| Testosterone | IVW | 0.96 (0.90, 1.05) | 0.463 | 1.06 (0.97, 1.15) | 0.214 | 0.96 (0.83, 1.10) | 0.547 | 1.03 (0.87, 1.22) | 0.731 | 1.07 (0.89, 1.27) | 0.479 | 0.88 (0.69, 1.11) | 0.269 |
| | MR-Egger | 0.97 (0.86, 1.09) | 0.644 | 1.04 (0.85, 1.26) | 0.713 | 0.88 (0.71, 1.10) | 0.270 | 1.13 (0.76, 1.69) | 0.549 | 0.81 (0.62, 1.08) | 0.152 | 0.68 (0.39, 1.18) | 0.169 |
| | Weighted median | 0.93 (0.83, 1.04) | 0.184 | 1.06 (0.94, 1.19) | 0.370 | 0.89 (0.72, 1.10) | 0.277 | 1.08 (0.84, 1.39) | 0.523 | 0.89 (0.67, 1.19) | 0.438 | 0.81 (0.57, 1.14) | 0.227 |
| | p for intercept | 1.00 (1.00, 1.00) | 0.998 | 1.00 (0.99, 1.01) | 0.854 | 1.00 (1.00, 1.01) | 0.348 | 1.00 (0.99, 1.01) | 0.615 | 1.01 (1.00, 1.02) | 0.017 | 1.01 (0.99, 1.03) | 0.314 |
| | MR-PRESSO | 0.97 (0.90, 1.05) | 0.464 | 1.06 (0.97, 1.15) | 0.216 | 0.96 (0.83, 1.10) | 0.549 | 1.03 (0.87, 1.22) | 0.732 | 1.07 (0.89, 1.27) | 0.478 | 0.88 (0.69, 1.11) | 0.270 |
| Estradiol | IVW | 0.99 (0.89, 1.11) | 0.923 | 0.95 (0.71, 1.26) | 0.724 | 0.98 (0.81, 1.18) | 0.826 | 1.04 (0.63, 1.73) | 0.873 | 0.90 (0.71, 1.15) | 0.403 | 1.39 (0.74, 7.15) | 0.310 |
| | MR-Egger | 1.00 (0.73, 1.36) | 0.993 | 0.89 (0.59, 1.34) | 0.598 | 0.93 (0.52, 1.67) | 0.812 | 1.15 (0.56, 2.34) | 0.719 | 0.61 (0.29, 6.15) | 0.233 | 1.76 (0.74, 3.15) | 0.234 |
| | Weighted median | 1.05 (0.92, 1.20) | 0.432 | 0.95 (0.68, 1.32) | 0.745 | 0.93 (0.74, 1.16) | 0.508 | 1.32 (0.67, 2.57) | 0.422 | 0.88 (0.65, 1.15) | 0.411 | 1.96 (0.81, 5.15) | 0.135 |
| | p for intercept | 1.00 (0.96, 1.04) | 0.980 | 1.00 (0.99, 1.02) | 0.669 | 1.01 (0.94, 1.08) | 0.856 | 0.99 (0.96, 1.02) | 0.707 | 1.05 (0.96, 0.15) | 0.312 | 0.99 (0.95, 0.15) | 0.441 |
| | MR-PRESSO | 0.99 (0.89, 1.11) | 0.925 | 0.95 (0.71, 1.26) | 0.732 | 0.98 (0.81, 1.18) | 0.831 | 1.04 (0.63, 1.73) | 0.877 | 0.90 (0.71, 1.15) | 0.425 | 1.39 (0.74, 2.63) | 0.335 |

OR = odds ratio. CI = confidence interval. IVW = inverse variance weighting.

**Table 5.** Associations of serum E2 levels instrumented by rs7173595 in the CYP19A1 gene region with COVID-19 outcomes.

| Sex | Phenotype | beta | SE | OR (95% CI) | p Effect |
|---|---|---|---|---|---|
| | Susceptibility | −1.14 | 0.88 | 0.32 (0.06, 1.80) | 0.195 |
| | Hospitalization | −1.27 | 1.60 | 0.28 (0.01, 6.46) | 0.426 |
| Female | Severity | −1.49 | 2.06 | 0.22 (0.00, 12.73) | 0.469 |
| | Susceptibility | −1.00 | 0.77 | 0.37 (0.08, 1.67) | 0.195 |
| | Hospitalization | −1.11 | 1.40 | 0.33 (0.02, 5.11) | 0.426 |
| Male | Severity | −1.31 | 1.80 | 0.27 (0.01, 9.26) | 0.469 |

E2 = estradiol. OR = odds ratio. CI = confidence interval.

an important role in the repair of lung tissue by regulating the proliferation and differentiation of alveolar epithelial cells (AECs) (*Narasaraju et al., 2006*). Airway inflammation can be mitigated when apoptotic cells are engulfed by pulmonary epithelial cells (*Juncadella et al., 2013*). IGF-1 has also been shown to upregulate engulfment by professional phagocytes such as dendritic cells (*Xuan et al., 2017*), and inhibit IL-6 production from lipopolysaccharide-induced AECs (*Wang et al., 2019*). Both of these mechanisms are beneficial to the regression of local inflammation. Jakn et al. showed that IGF-1 binds to IGF-1 receptor (IGF-1R) on airway epithelial cells of non-professional phagocytic cells, which can promote the phagocytosis of microparticles by airway epithelial cells (*Han et al., 2016*). Transforming growth factor β1 derived from AECs activated alveolar macrophages (AMs) to secrete IGF-1 into the alveolar fluid in response to stimulation of the airway by inflammatory signals. This AM-derived IGF-1 attenuated the p38 mitogen-activated protein kinase inflammatory signal in AECs and promoted the phagocytosis of apoptotic cells by AECs. This two-way communication between AECs and AMs represents a well-tuned system for the regulation of the inflammatory response in alveoli (*Mu et al., 2020*). Taken together, these studies provide biological evidence supporting that IGF-1 might be an important anti-inflammatory factor in the alveolar microenvironment and thus may contribute to improve COVID-19 outcomes. More studies are required to determine whether novel therapeutic strategy targeting on IGF-1 pathway might improve COVID-19 prognosis.

IGF-1 level is regulated by estrogen and the functional interactions between estradiol and IGF-1 signaling system involve several transcriptional and posttranscriptional mechanisms. Specifically, IGF-1 can affect estrogen receptor α action by enhancing its expression and potentiating its transcriptional activity in a ligand-independent manner (*Lange, 2004*; *Edwards et al., 1993*; *Shupnik, 2004*). On the other hand, E2 can enhance IGF-1 signaling by upregulating the expression of IGF-1 (*Umayahara et al., 1994*), IGF-1R (*Bartucci et al., 2001*), and some IGF-1-binding proteins (*Qin et al., 1999*). This may explain the same direction from the IVW analysis of IGF-1, estradiol, and COVID-19 outcomes. Estrogen is found to have immune enhancing effect (*Taneja, 2018*) to trigger the local immune response by activating a plethora of cells such as phagocytes, dendritic cells, natural killers, and CD8[+] T cells. Once these immune cells are activated, they could fight against the infection by destroying the virus and thus preventing its diffusion to the lower respiratory tract or by decreasing the viral load. Experimental tests have also reported that estradiol can affect angiotensin-converting enzyme 2 and FURIN expression, with the potential of mitigating SARS-CoV-2 infection (*Glinsky, 2020*). However, our study did not find any supportive evidence for the associations between estradiol and COVID-19, which might be due to the small variance of estradiol explained by genetic instruments.

Our studies showed that SHBG or testosterone may not be associated with COVID-19 outcomes, which is consistent with the research findings of *Liu et al., 2022*. They also observed a null causal relationship for testosterone or SHBG levels with COVID-19 outcomes in females and males. Meanwhile, epidemiologic data (*Peckham et al., 2020*) indicate that while men are not more predisposed to contracting COVID-19, they are more likely to develop severe illness following the infection compared with women. However, our study observed null causal relationship for testosterone levels with COVID-19 outcomes in both females and males. According to the available evidence on the role of testosterone in COVID-19, it appears that both high and low testosterone levels can be associated with poor COVID-19 outcomes (*Ho et al., 2022*). A study demonstrated androgen deprivation

**Table 6.** Testosterone, SHBG, IGF-1, and COVID-19 outcomes in Mendelian randomization (MR) analyses adjusting BMI.

| Exposure | Method | Susceptibility | | | | | Hospitalization | | | | | Severity | | | | |
|---|---|---|---|---|---|---|---|---|---|---|---|---|---|---|---|---|
| | | SNPs | OR (95% CI) | P Effect | P Heterogeneity | P Intercept | SNPs | OR (95% CI) | P Effect | P Heterogeneity | P Intercept | SNPs | OR (95% CI) | P Effect | P Heterogeneity | P Intercept |
| Testosterone | IVW | | 0.95 (0.83,1.07) | 0.386 | 0.006 | – | | 0.83 (0.64,1.06) | 0.134 | 0.041 | – | | 0.84 (0.60,1.17) | 0.304 | 0.030 | – |
| | MR-Egger | | 0.93 (0.77,1.13) | 0.484 | 0.006 | 0.855 | | 0.83 (0.56,1.21) | 0.324 | 0.038 | 0.991 | | 0.83 (0.50,1.37) | 0.466 | 0.027 | 0.949 |
| | Weighted median | | 0.90 (0.72,1.12) | 0.331 | – | – | | 0.82 (0.52,1.28) | 0.375 | – | – | | 0.71 (0.42,1.21) | 0.214 | – | – |
| | Simple mode | | 1.13 (0.70,1.82) | 0.610 | – | – | | 0.68 (0.24,1.91) | 0.465 | – | – | | 0.37 (0.07,1.88) | 0.229 | – | – |
| | Weighted mode | | 0.95 (0.79,1.13) | 0.540 | – | – | | 0.81 (0.56,1.17) | 0.273 | – | – | | 0.65 (0.40,1.06) | 0.085 | – | – |
| | MR-PRESSO | 306 | 0.94 (0.83,1.07) | – | – | – | 294 | 0.83 (0.64,1.06) | – | – | – | 307 | 0.83 (0.64,1.06) | – | – | – |
| SHBG | IVW | | 0.90 (0.79,1.04) | 0.160 | 0.002 | – | | 0.84 (0.64,1.10) | 0.209 | 0.047 | – | | 0.89 (0.62,1.26) | 0.511 | 0.058 | – |
| | MR-Egger | | 0.94 (0.76,1.15) | 0.538 | 0.001 | 0.663 | | 0.81 (0.54,1.21) | 0.299 | 0.043 | 0.794 | | 0.89 (0.53,1.49) | 0.666 | 0.054 | 0.978 |
| | Weighted median | | 0.90 (0.71,1.13) | 0.356 | – | – | | 0.81 (0.52,1.28) | 0.377 | – | – | | 0.72 (0.42,1.23) | 0.230 | – | – |
| | Simple mode | | 1.05 (0.60,1.84) | 0.860 | – | – | | 1.25 (0.42,3.78) | 0.689 | – | – | | 0.97 (0.22,4.22) | 0.967 | – | – |
| | Weighted mode | | 0.94 (0.77,1.15) | 0.570 | – | – | | 0.81 (0.55,1.20) | 0.295 | – | – | | 0.72 (0.43,1.22) | 0.224 | – | – |
| | MR-PRESSO | 308 | 0.90 (0.79,1.04) | – | – | – | 198 | 0.84 (0.64,1.10) | – | – | – | 309 | 0.89 (0.62,1.26) | – | – | – |
| IGF-1 | IVW | | 0.76 (0.60,0.96) | 0.021 | 0.172 | – | | 0.61 (0.41,0.90) | 0.014 | 0.688 | – | | 0.84 (0.52,1.38) | 0.497 | 0.534 | – |
| | MR-Egger | | 0.88 (0.58,1.33) | 0.554 | 0.168 | 0.390 | | 0.77 (0.39,1.50) | 0.458 | 0.676 | 0.403 | | 1.55 (0.71,3.39) | 0.284 | 0.757 | – |
| | Weighted median | | 0.75 (0.57,0.99) | 0.046 | – | – | | 0.75 (0.45,1.24) | 0.260 | – | – | | 0.75 (0.38,1.48) | 0.410 | – | – |
| | Simple mode | | 0.65 (0.38,1.11) | 0.135 | – | – | | 0.64 (0.30,1.37) | 0.265 | – | – | | 0.75 (0.25,2.31) | 0.629 | – | – |
| | Weighted mode | | 0.76 (0.56,1.03) | 0.096 | – | – | | 0.71 (0.44,1.15) | 0.185 | – | – | | 0.72 (0.36,1.47) | 0.383 | – | – |
| | MR-PRESSO | 15 | 0.76 (0.60,0.96) | – | – | – | 15 | 0.61 (0.43,0.86) | – | – | – | 17 | 0.84 (0.53,1.35) | – | – | – |

SNP = single-nucleotide polymorphism. OR = odds ratio. CI = confidence interval. IVW = inverse variance weighting. SHBG = sex hormones-binding globulin. IGF-1 = insulin-like growth factor-1.

therapy (ADT) exposure was associated with a reduction in COVID-19 severity (*Lee et al., 2022*). By contrast, the Ohio study did not identify any protective effect of ADT on the severity of COVID-19 outcomes (*Klein et al., 2021*). Androgen-related treatments showed that transmembrane serine protease 2 (TMPRSS2) expression and SARS-CoV-2 entry in human lung cells have been reduced by antiandrogens (*Leach et al., 2021*; *Deng et al., 2021*; *Qiao et al., 2020*). Additionally, androgens have numerous immunosuppressive effects such as decreasing proinflammatory cytokine release (e.g., IFNγ and TNF) or increasing anti-inflammatory cytokine release (e.g., IL-4 and IL-10), reducing T helper 1 (Th1) and T helper 17 (Th17) cell differentiation, inducing Treg differentiation and regulating B-cell development (*Olsen and Kovacs, 2011*; *Henze et al., 2020*; *Trigunaite et al., 2015*). Paradoxically, these immunosuppressive effects of testosterone might be beneficial to overcome the heightened inflammatory environment that predisposes to severe COVID-19. Recent research has revealed that males with COVID-19 have lower testosterone levels (*Ma et al., 2021*). Another study found a negative association between total testosterone levels and biochemical markers of COVID-19 severity (*Rastrelli et al., 2021*). Lower testosterone concentrations were associated with higher concentrations of IL-6, CRP, IL-1 receptor antagonist, hepatocyte growth factor, and IFNγ-inducible protein 10 (*Dhindsa et al., 2021*). Therefore, additional research efforts need to be made to investigate the complex relationships furtherly.

The major advantage of our study is the design taking the advantages of MR approach and used several sensitivity analyses to test the robustness of the MR findings. The application of MR analysis reduces the influence of confounding factors and reverse causality so that reliable causal estimations were obtained to complement the observational findings. The potential limitations of this study also need to be acknowledged. Our study may suffer from weak instrument bias, especially within sensitivity analyses that restricted to smaller sets of genetic instruments. In TSMR, this bias would tend to make estimates closer to the null. Since there is no available data on recovery status for COVID-19 patients in UK Biobank, the current study did not take recovery as a potential competing risk into account. We could not assess the sex-specific associations in IGF-1 and COVID-19 due to no data by sex in HGI. Moreover, the MR was merely based on individuals of European ancestry. Our findings might not be generalized to other populations. It should also be noted that the study findings are based on evidence from genetic data, additional large and prospective cohort studies with available IGF-1 data and information on COVID-19 susceptibility and clinical outcomes are needed to validate the findings.

In conclusion, our study indicated that genetically predicted high IGF-1 levels were associated with decrease the risk of COVID-19 susceptibility and hospitalization, but these associations did not survive the Bonferroni correction of multiple testing. Further studies are needed to validate the findings and explore whether IGF-1 could be a potential intervention target to reduce COVID-19 risk.

## Data availability statement

Data analyzed in the present study are GWAS summary statistics, which have been made publicly available. GWAS summary level data of COVID-19 HGI could be downloaded from https://www.covid19hg.org/results/. GWAS summary level data of sex hormones and IGF-1 in UK Biobank could be downloaded from GWAS catalog. All genome-wide significant SNPs have been provided in *Supplementary file 1a–d*. All analyses were performed using R statistical package freely available at https://cran.r-project.org/mirrors.html. The TSMR package is available at https://mrcieu.github.io/TwoSampleMR/(*Hemani et al., 2020*).

## Acknowledgements

The authors are thankful for all the participants that contributed to the UK Biobank study. Funding Statement: The funders had no role in study design, data collection, and interpretation, or the decision to submit the work for publication Funding information: This paper was supported by the following grants: Natural Science Fund for Distinguished Young Scholars of Zhejiang Province (LR22H260001) to Xue Li. CRUK Career Development Fellowship(C31250/A22804) to Evropi Theodoratou, the Swedish Heart Lung Foundation (Hjärt-Lungfonden, 20210351), the Swedish Research Council (Vetenskapsrådet, 2019-00977), and the Swedish Cancer Society (Cancerfonden) to Susanna C Larsson.

# Additional information

## Funding

| Funder | Grant reference number | Author |
| --- | --- | --- |
| Natural Science Foundation of Zhejiang Province | Distinguished Young Scholars | Xue Li |
| Cancer Research UK | CRUK Career Development Fellowship | Evropi Theodoratou |
| Swedish Cancer Foundation | | Susanna C Larsson |
| Swedish Research Council | | Susanna C Larsson |
| Swedish Heart Lung Foundation | | Susanna C Larsson |
| Science Fund for Distinguished Young Scholars of Zhejiang Province | | Susanna C Larsson |

The funders had no role in study design, data collection and interpretation, or the decision to submit the work for publication.

## Author contributions

Xinxuan Li, Data curation, Software, Formal analysis, Validation, Investigation, Visualization, Methodology, Writing - original draft; Yajing Zhou, Formal analysis, Writing - original draft; Shuai Yuan, Writing - original draft, Writing – review and editing; Xuan Zhou, Resources, Formal analysis; Lijuan Wang, Formal analysis; Jing Sun, Validation, Methodology; Lili Yu, Resources, Validation; Jinghan Zhu, Visualization; Han Zhang, Validation; Nan Yang, Resources, Methodology; Shuhui Dai, Resources; Peige Song, Conceptualization, Writing – review and editing; Susanna C Larsson, Funding acquisition, Writing – review and editing; Evropi Theodoratou, Supervision, Funding acquisition, Writing – review and editing; Yimin Zhu, Conceptualization, Methodology, Writing – review and editing; Xue Li, Conceptualization, Data curation, Supervision, Funding acquisition, Project administration, Writing – review and editing

## Author ORCIDs

Xinxuan Li http://orcid.org/0000-0001-8922-661X
Xue Li http://orcid.org/0000-0001-6880-2577

## Ethics

UK Biobank received ethical approval from the North West Multi-centre Research Ethics Committee, the National Information Governance Board for Health and Social Care in England and Wales, and the Community Health Index Advisory Group in Scotland. All participants provided written informed consent. All institutions contributing cohorts to the COVID-19 HGI received ethics approval from their respective research ethics review boards. The ethical permit for MR analyses based on summary-level data was unnecessary.

## Decision letter and Author response

Decision letter https://doi.org/10.7554/eLife.79720.sa1
Author response https://doi.org/10.7554/eLife.79720.sa2

# Additional files

## Supplementary files
- MDAR checklist
- Supplementary file 1. Genetic instruments for testosterone, estrogen, SHBG, and IGF-1.
- Reporting standard 1. Reporting checklist for case-control study.

## Data availability

Data analysed in the present study are GWAS summary statistics, which have been made publicly available. GWAS summary level data of COVID-19-HGI could be downloaded from https://www.covid19hg.org/results/. GWAS summary level data of sex hormones and IGF-1 in UK biobank could be downloaded from GWAS catalog (http://ftp.ebi.ac.uk/pub/databases/gwas/summary_statistics/GCST90019001-GCST90020000/). All genome-wide significant SNPs have been provided in Supplementary Tables 1 to 4 in Supplementary file 1. All analyses were performed using R statistical package freely available at https://cran.r-project.org/mirrors.html. The Two-sample MR package is available at https://mrcieu.github.io/TwoSampleMR/.

The following previously published dataset was used:

| Author(s) | Year | Dataset title | Dataset URL | Database and Identifier |
|---|---|---|---|---|
| COVID-19 Host Genetics Initiative. | 2021 | COVID-19 Host Genetics Initiative round 5 | https://www.covid19hg.org/results/ | COVID-19 HGI, covid19hg |

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
