## [Editor Report]

Using publicly available genetic data, Li and colleagues tested the association and inferred the causality of genetic variants predicted to alter the levels of testosterone, estrogen, SHBG, or IGF-1, against susceptibility, severity and outcome of SARS-Cov2 infection. The main strength of the study is the large cohort which adds to the robustness of the data.

---

## [Decision Letter]

**Decision letter after peer review:**

Thank you for submitting your article "Genetically predicted high IGF-1 levels showed protective effects on COVID-19 susceptibility and hospitalization: A Mendelian Randomisation study with data from 60 studies across 25 countries" for consideration by *eLife*. Your article has been reviewed by 3 peer reviewers, including Evangelos J Giamarellos-Bourboulis as Reviewing Editor and Reviewer #1, and the evaluation has been overseen by Jos van der Meer as the Senior Editor.

Recommendations for the authors:

– The main limitation of the study is the lack of any real-world validation cohort where IGF-1 is measured.

– Another limitation is the lack of any transcriptomic data to support the findings. Can the authors describe any eQTLs to recompensate for this?

– The authors leveraged previous reports of genetic variants in association with testosterone, estrogen, SHBG, and IGF-1 levels, outside of COVID-19 cohorts. Testosterone, estrogen, SHBG, and IGF-1 levels were not measured specifically in the context of the COVID-19 Host Genetics Initiative. Secondly, no attempts were made to further validate, either experimentally or in other external cohorts, their findings related to IGF-1 levels. Thirdly, the statistics is seemingly lacking appropriate control of the false discovery rate. Although the study is a targeted meta-analysis, by examining 657 different markers one is bound to identify an association by chance. In fact, applying a Bonferroni correction to the reported p-values for IGF-1 related SNPs yields non-significant probabilities. Fourth, the methods section lacks important details, for example handling of the known "winner's curse" bias in inverse variance weighting mendelian randomization, thereby decreasing the potential for reproducibility by other groups. Lastly, the methods employed did not adequately address potential data confounders, for example treatment of patients with dexamethasone or the influence of BMI (inferred in other studies as a causal risk factor) on model estimates. Against this backdrop, the interpretation of the findings by Li and colleagues is complicated and conclusions related to genetic proxies of IGF-1 levels are unreliable.

– Based on the cited references (13 and 14), selection of SNPs in the meta-analysis is subject to a bias towards non-infectious conditions. It is important to consider that levels of testosterone, estradiol, SHBG, and IGF-1 could be substantially altered during the course of the infection, either in the acute phase or later stages. What are the levels of testosterone, estradiol, SHBG and IGF-1 in the context of viral infections, as well as crucially in COVID-19? How do those levels influence the association of the selected SNPs?

– Throughout the results, particularly in Figure 2, confidence intervals are large and indicates a potential complication due to unaddressed confounders or subgroups within the cohort. How many of the COVID-19 patients were administered corticosteroid therapy? How many were administered tocilizumab? How does treatment effect the model estimates? The authors should provide more characteristics of the COVID-19 patients with a more extensive investigation on the potential confounders and treatment effects on MR estimates. How does BMI influence model estimates and results?

– Inverse-variance weighted two-sample Mendelian randomization is indeed the most extensively used method utilizing GWAS summary stats to infer on causality of the exposure (ie. SNP variant) and outcome variables. The approach suffers from various biases that were seemingly not handled in the study herein described, for example, the winner's curse and potential pleiotropy. The authors should provide a more objective assessment of potential biases and address them accordingly.

– Notwithstanding the large sample size of the discovery cohort obtained from the host genetics in COVID-19 initiative, it is imperative to expand on the findings by ascertaining the robustness of the IFG-1 related SNPs. Do the IGF-1-related SNPs influence IGF-1 levels in the context of a viral infection? Preferably of course in COVID-19. Also, do the SNPs constitute eQTLs for IGF-1 expression in primary organs, particularly lungs, liver, and kidneys?

– The methods section needs to be expanded substantially by the addition of important details. For example, how was LD clumping performed? Using PLINK? Why was an r2 cutoff of < 0.01 used? Why not r2 < 0.001?

– The sentence on page 5 line 38 reading "postmenopausal women (22) and men(13, 23)." needs correction as it implies men were also postmenopausal.

– Page 7, line 196, the statement that reads "The cytokine storm related respiratory distress syndrome…" should be revised, avoiding the sensationalistic "cytokine storm" description of what has been reported elsewhere as merely a cytokine breeze at best.

– The Discussion on cytokine storm is arbitrary and should be omitted. The authors do not provide any evidence on the association of IGF-1 with the cytokine storm.

– This is another trial finding no association of sex hormones to COVID-19. This has to be discussed more extensively in the discussion; now discussion is more focused on IGF (the positive finding) but negative results should be put into context. Please refer also to PMID: 35577073, PMID: 35470422, PMID: 35602518

[Editors’ note: further revisions were suggested prior to acceptance, as described below.]

Thank you for resubmitting your work entitled "Genetically predicted high IGF-1 levels showed protective effects on COVID-19 susceptibility and hospitalization: A Mendelian Randomisation study with data from 60 studies across 25 countries" for further consideration by *eLife*. Your revised article has been evaluated by Jos van der Meer (Senior Editor) and a Reviewing Editor.

The manuscript has been improved but there are some remaining issues that need to be addressed, as outlined below:

– The study hinges solely on MR analysis of selected SNPs used as proxies of testosterone, estradiol, SHBG, and IGF-1 levels. None of the markers were directly measured, and no information on their levels in the context of infections or COVID-19 was given. The authors argue at length that measuring the markers in the specific cohort is of limited value because their genetic causality modelling approach supersedes the need for such measurements. While MR models have indeed emerged as interesting tools to infer causality, it is also important to provide a potential mechanism linking genetic variants to the phenotype. Without direct measurements of the said markers minimize the potential to provide readers with a possible mechanism bridging genetic variant and phenotype.

– The number of tests performed in the study is equivalent to the number of SNPs that were investigated, that is, 657. Hence, the multiple-testing correction must take into account 657 tests. The authors are wrong in adjusting their probabilities just for the 4 markers, which is more attuned to p-value hacking. Therefore, my initial comments on the lack of significant findings after considering multiple-test corrections remain a concern. Hence, the conclusions pertaining to IGF-1 variants do not reflect on the author's findings. Based on this result, I suggest being more objective by stating that none of the genetic variants can be deemed as a causal variant.

---

## [Author Response]

Recommendations for the authors:– The main limitation of the study is the lack of any real-world validation cohort where IGF-1 is measured.

Thanks for this insightful suggestion. We agree with the reviewer that cohort study is a straightforward and effective way to validate whether the IGF-1 levels are associated with the risk of COVID-19 outcomes. There is one observational study that demonstrated an inverse association between measured circulating levels of IGF-1 and COVID-19 mortality risk among COVID-19 patients in the UK biobank (1), however, the causality of the association remains unestablished due to the potential limitations of this study (e.g., residual confounding and reverse causality) and the shortcoming that the blood samples were taken over a decade ago in UK biobank and are unlikely to be representative of participants’ IGF-1 levels at the time of the pandemic. We therefore applied the MR design as an alternative approach to explore this association, which has several advantages. By employing genetic variants as instrumental variables to proxy the life-long time exposure, the MR approach can strengthen causal inference by minimizing unobserved confounding and diminishing reverse causality, since genetic variants are randomly assorted at conception, and thus unrelated to common confounding factors and unaffected by the onset and progression of the disease. We now highlight this proposed point as future research efforts in the Discussion section.

Discussion, page 11, line 289-291: “It should also be noted that our study findings are based on evidence from genetic data, additional large and prospective cohort studies with available IGF-1 data and information on COVID-19 susceptibility and clinical outcomes are needed to validate the findings.”

– Another limitation is the lack of any transcriptomic data to support the findings. Can the authors describe any eQTLs to recompensate for this?

Many thanks for your constructive suggestion. We agree with the reviewer that the lack of any transcriptomic data is a limitation. As suggested, we searched 157 significant single-tissue eQTLs of IGF-1 (ENSG00000017427.15) in all tissues by GTEs and identified 2 eQTLs of IGF-1 in small intestine-terminal ileum, 9 in spleen, 15 in muscle skeletal and 131 in testis, whilst we didn’t find any eQTLs of IGF-1 in blood cells or lung tissue. Considering the transcriptomic effect is different across tissues, we are therefore on the side of caution to use the eQTLs in other tissues to perform analysis.

– The authors leveraged previous reports of genetic variants in association with testosterone, estrogen, SHBG, and IGF-1 levels, outside of COVID-19 cohorts. Testosterone, estrogen, SHBG, and IGF-1 levels were not measured specifically in the context of the COVID-19 Host Genetics Initiative. Secondly, no attempts were made to further validate, either experimentally or in other external cohorts, their findings related to IGF-1 levels. Thirdly, the statistics is seemingly lacking appropriate control of the false discovery rate. Although the study is a targeted meta-analysis, by examining 657 different markers one is bound to identify an association by chance. In fact, applying a Bonferroni correction to the reported p-values for IGF-1 related SNPs yields non-significant probabilities. Fourth, the methods section lacks important details, for example handling of the known "winner's curse" bias in inverse variance weighting mendelian randomization, thereby decreasing the potential for reproducibility by other groups. Lastly, the methods employed did not adequately address potential data confounders, for example treatment of patients with dexamethasone or the influence of BMI (inferred in other studies as a causal risk factor) on model estimates. Against this backdrop, the interpretation of the findings by Li and colleagues is complicated and conclusions related to genetic proxies of IGF-1 levels are unreliable.

Many thanks for these helpful comments. Firstly, we agree with reviewer that testosterone, estrogen, SHBG, and IGF-1 levels were not measured specifically in the context of the COVID-19 Host Genetics Initiative. But it does not affect our MR analysis of causality since genetic alleles associated with the exposure are randomly assorted at conception and thus unrelated to self-selected lifestyle and environmental factors, and are not modified by disease either. Mendelian randomization (MR) study, which employs genetic variants as unbiased proxies for the risk factor, is a genetic method that can strengthen the inference on the causal nature of exposure-outcome associations by diminishing the likelihood of confounding and eliminating reverse causality in conventional observational studies (2, 3).

Secondly, we approve of the reviewer that it’s better to further validate our findings. MR study use genetic variants as instruments, which are not associated with a wide range of behavioral, social and physiological factors that confound associations with outcomes. On the other hand, they can serve as unconfounded indicators of particular trait values. Genetic variants and their effects are subject to relatively little measurement error or bias.(4) We agree it should be acknowledged as a limitation or future research effort that additional large and prospective cohort studies with available IGF-1 data and information on COVID-19 susceptibility and clinical outcomes are needed to validate the findings.

Thirdly, we totally used 657 SNPs, including 315 for testosterone, 319 for SHBG, 7 for estradiol and 16 for IGF-1. The 16 SNPs were analyzed as a whole for IGF-1. We apologize for not considering correction. When applying the Bonferroni correction for multiple testing, the adjusted P value < 0.05/4 = 0.013 (four biomarkers) should be considered as statistically significant. As suggested, we have now added Bonferroni correction in the Materials and methods section as below. When adopting the adjusted P-value threshold, we observed marginally significant association between IGF-1 and COVID-19 hospitalization with a p-value of 0.018. Any association with P-value<0.05 is now described as nominally significant, throughout the main text.

Lastly, we agree with the reviewer that addressing potential data confounders is inadequate in our study. We therefore used PhenoScanner to find out all reported associations for each SNP and exclude several SNPs to be associated with body mass index in the subsequent analysis additionally. Besides, we did not find any SNP associated with dexamethasone therapy at the threshold of 5×10^-8^ in European ancestry samples, as examined by the PhenoScanner. As suggested by the reviewer, we conduct the sensitive analysis and add the results in Table 6. Revisions in the manuscript are made as below.

Materials and methods section, page 5, line 155-159: “To control potential data confounder, we selected SNPs associated with testosterone, estrogen, SHBG, and IGF-1 only, excluding SNPs associated with BMI which is thought to be a causal risk factor for COVID-19(5) at the threshold of 5×10^-8^ in European ancestry samples by querying PhenoScanner. SNPs in estrogen were not exclude for their irrelevance to BMI.”

Results section, page 7, line 167-169: “Associations of IGF-1 levels with COVID-19 susceptibility and hospitalization were not statistically significant after Bonferroni correction, albeit showing a nominal significance at P<0.05.”

Results section, page 7, line 194-200: “As shown in Table 6, after removing SNPs associated with BMI, we found similar associations of genetically predicted IGF-1 levels with the risk of COVID-19 susceptibility (OR=0.76; 95%CI: 0.60-0.96, P=0.021), hospitalization (OR=0.61; 95%CI: 0.41-0.90, P=0.014), risk of severity (OR=0.84; 95%CI: 0.52-1.38, P=0.497) in which we detected no moderate heterogeneity, and no indication of horizontal pleiotropy in MR-Egger, and no outlier in MR-PRESSO analyses. No causal associations of genetically predicted testosterone and SHBG with COVID-19 were found, but the directions were consistent with results in Table 2.”

– Based on the cited references (13 and 14), selection of SNPs in the meta-analysis is subject to a bias towards non-infectious conditions. It is important to consider that levels of testosterone, estradiol, SHBG, and IGF-1 could be substantially altered during the course of the infection, either in the acute phase or later stages. What are the levels of testosterone, estradiol, SHBG and IGF-1 in the context of viral infections, as well as crucially in COVID-19? How do those levels influence the association of the selected SNPs?

Many thanks for this comment. Agreeing with the reviewer that these SNPs were selected from non-infectious conditions, in which case, the genetic alleles associated with the exposure are randomly assorted at conception and thus unrelated to self-selected lifestyle and environmental factors and are not modified by disease either; in other word, disease conditions would not influence the selection of SNPs. We acknowledge that the levels of testosterone, estradiol, SHBG, and IGF-1 might be changed during the course of infection, which means that the association of interest would be easier to suffer from reverse causality. Given the fact that we were unbale to obtain the measured levels of biomarkers during the course of infection, we therefore alternatively used the genetically predicted levels to proxy the lifelong-time exposure, which was believed to be more persistent than a single measurement of these biomarkers and to minimize the potential of reverse causation.

– Throughout the results, particularly in Figure 2, confidence intervals are large and indicates a potential complication due to unaddressed confounders or subgroups within the cohort. How many of the COVID-19 patients were administered corticosteroid therapy? How many were administered tocilizumab? How does treatment effect the model estimates? The authors should provide more characteristics of the COVID-19 patients with a more extensive investigation on the potential confounders and treatment effects on MR estimates. How does BMI influence model estimates and results?

Thank you for this comment. This study was designed as Two-sample MR based on the GWAS summary data, with the advantage to resist the common confounders. As explained above, MR study use genetic variants as instruments, which are not associated with a wide range of behavioral, social and therapeutic factors that might confound associations with the outcome. Therefore, whether the patient has received medication does not affect our analysis of the relationship between IGF-1 and COVID-19. However, as proposed by the reviewer as an example, if the used genetic IVs have any pleiotropic effect on BMI, this will violate the MR assumptions. To address this point, we have added more sensitivity analyses in our study. The revisions have been made as below.

Materials and methods section, page 5, line 155-159: “To control potential data confounder, we selected SNPs associated with testosterone, estrogen, SHBG, and IGF-1 only, excluding SNPs associated with BMI which is thought to be a causal risk factor for COVID-19(5) at the threshold of 5×10^-8^ in European ancestry samples by querying PhenoScanner.(6) SNPs in estrogen were not exclude for their irrelevance to BMI.”

Results section, page 7, line194-200: “As shown in Table 6, after removing SNPs associated with BMI, we found similar associations of genetically predicted IGF-1 levels with the risk of COVID-19 susceptibility (OR=0.76; 95%CI: 0.60-0.96, P=0.021), hospitalization (OR=0.61; 95%CI: 0.41-0.90, P=0.014), risk of severity (OR=0.84; 95%CI: 0.52-1.38, P=0.497) in which we detected no moderate heterogeneity, and no indication of horizontal pleiotropy in MR-Egger, and no outlier in MR-PRESSO analyses. No causal associations of genetically predicted testosterone and SHBG with COVID-19 were found, but the directions were consistent with results in Table 2.”

– Inverse-variance weighted two-sample Mendelian randomization is indeed the most extensively used method utilizing GWAS summary stats to infer on causality of the exposure (ie. SNP variant) and outcome variables. The approach suffers from various biases that were seemingly not handled in the study herein described, for example, the winner's curse and potential pleiotropy. The authors should provide a more objective assessment of potential biases and address them accordingly.

Many thanks for this helpful comment. We agree with the reviewer that the MR approach might suffer from various biases, we therefore performed a series of sensitivity analysis to examine the presence of pleiotropy and test the robustness of the MR results. In our study, MR-Egger regression was used to observe and correct potential directional pleiotropy, which was assessed by its intercept test. The weighted median method was performed to provide unbiased causal estimates if at least 50% instrumental variables were valid. MR-PRESSO method can detect SNP outliers and estimate the association after removal of these outliers. The differences in estimates between before and after outlier removal were examined by the embedded distortion test. Cochrane’s Q value was used to assess the heterogeneity among estimates of genetic instruments. As a result, we detected no moderate heterogeneity, and no indication of horizontal pleiotropy in MR-Egger, and no outlier in MR-PRESSO analyses. The influence of winner's curse was considered when selecting data for two-sample MR analysis, for which we used the modified version of Host Genetics Initiative (COVID-19-HGI) (Round 5, European population) where UKB data were excluded to avoid any overlapped samples between the exposure and the outcome GWAS datasets.

– Notwithstanding the large sample size of the discovery cohort obtained from the host genetics in COVID-19 initiative, it is imperative to expand on the findings by ascertaining the robustness of the IFG-1 related SNPs. Do the IGF-1-related SNPs influence IGF-1 levels in the context of a viral infection? Preferably of course in COVID-19. Also, do the SNPs constitute eQTLs for IGF-1 expression in primary organs, particularly lungs, liver, and kidneys?

Thanks for your insightful comment. Based on the Mendelian genetic law, MR refers to the use of genetic variants to develop causal inferences from observational data, if the variant genotype is associated with the phenotype and the variant genotype associated with the risk of disease of interest through the phenotype. Thus, whether IGF-1-related SNPs influence IGF-1 levels in the context of a viral infection, it would not affect the application of MR approach for the inference of causality. As suggested, we searched 157 significant single-tissue eQTLs of IGF-1 (ENSG00000017427.15) in all tissues by GTEs and identified 2 eQTLs of IGF-1 in small intestine-terminal ileum, 9 in spleen, 15 in muscle skeletal and 131 in testis, whilst we didn’t find any eQTLs of IGF-1 in blood cells or lung tissue. Considering the transcriptomic effect of eQTLs is differential across tissues, we are therefore on the side of caution to use the eQTLs in other tissues to perform analysis.

– The methods section needs to be expanded substantially by the addition of important details. For example, how was LD clumping performed? Using PLINK? Why was an r2 cutoff of < 0.01 used? Why not r2 < 0.001?

Many thanks for the useful comment. We agree with the reviewer that the Methods section could be improved by providing more methodological details. In this study, all statistical analyses were two-sided and performed in R 4.0.4 software using the R package Two Sample MR and MR-PRESSO.(6) Therefore we performed LD clumping using not PLINK but R package “TwoSampleMR”. It should be acknowledged that with the decrease of r^2^ cutoff and the increase of kb cutoff, the less the numbers of IVs left which result in the less confounding and pleiotropy, but the corresponding statistical effect is lower; While the increase of IV numbers can improve the statistical effectiveness, it will also bring more bias. We therefore chose the r^2^ cutoff of < 0.01 which we thought is moderate. Moreover, given the confounding and pleiotropy, we used the Cochrane’s Q value to assess the heterogeneity among estimates of genetic instruments and the p value for intercept in MR-Egger to detect horizontal pleiotropy. Revisions in the manuscript are made as below.

Materials and methods section, page 4, line 101-110: “We restricted the analysis to SNPs in linkage equilibrium which were identified in the relevant GWAS at P<5 × 10^−8^ clumped on r^2^ = 0.01 within 10,000 kb using the 1000 genomes reference panel(7) to ensure sufficient statistical effectiveness. Among those pairs of SNPs that had LD r^2^ above the specified threshold (r^2^ = 0.01) only the SNP with the lower P value would be retained. SNPs absent from the LD reference panel were also removed. To test whether there was a weak instrumental variable bias, namely genetic variants selected as instrumental variables had a weak association with exposure, we calculated the F statistic if it is much greater than 10 for the instrument-exposure association, the possibility of weak instrumental variable bias is small. These analyses were conducted using the R package “TwoSampleMR”.(6)”

– The sentence on page 5 line 38 reading "postmenopausal women (22) and men(13, 23)." needs correction as it implies men were also postmenopausal.

Thank you for this careful comment. We have now revised it as “men (13,22) and postmenopausal women (23)” to be clear.

– Page 7, line 196, the statement that reads "The cytokine storm related respiratory distress syndrome…" should be revised, avoiding the sensationalistic "cytokine storm" description of what has been reported elsewhere as merely a cytokine breeze at best.

Many thanks for this insightful comment. Combined with comment 10 and comment 11, we agree with the reviewer that description and discussion on cytokine storm are inappropriate. As suggested, we have now removed the part of cytokine storm from manuscript.

– The Discussion on cytokine storm is arbitrary and should be omitted. The authors do not provide any evidence on the association of IGF-1 with the cytokine storm.

Many thanks for this helpful comment. We agree with the reviewer that discussion on the association of IGF-1 with the cytokine storm is not well-grounded. As suggested, we have now removed the discussion on cytokine storm.

– This is another trial finding no association of sex hormones to COVID-19. This has to be discussed more extensively in the discussion; now discussion is more focused on IGF (the positive finding) but negative results should be put into context. Please refer also to PMID: 35577073, PMID: 35470422, PMID: 35602518

Thank you very much for this comment. We agree with the reviewer that negative results should also be included in the discussion. We thank the reviewer for providing these important references. After carefully reading through the referred papers and other relevant articles, we have now discussed the negative results in the Discussion section. Revisions are made as below to extend our Discussion section.

Discussion section, page 10, line 253-277: “Our study showed that SHBG or testosterone may not be associated with COVID-19 outcomes, which is consistent with the research findings of Liu et al.(8) They also observed a null causal relationship for testosterone or SHBG levels with COVID-19 outcomes in females and males. Meanwhile, epidemiologic data (9) indicate that while men are not more predisposed to contracting COVID-19, they are more likely to develop severe illness following the infection compared with women. However, our study observed null causal relationship for testosterone levels with COVID-19 outcomes in both females and males. According to the available evidence on the role of testosterone in COVID-19, it appears that both high and low testosterone levels can be associated with poor COVID-19 outcomes.(10) A study demonstrated androgen deprivation therapy (ADT) exposure was associated with a reduction in COVID-19 severity.(11) By contrast, the Ohio study did not identify any protective effect of ADT on the severity of COVID-19 outcomes.(12) Androgen-related treatments showed that transmembrane serine protease 2 (TMPRSS2) expression and SARS-CoV-2 entry in human lung cells have been reduced by antiandrogens.(13-15) Additionally, androgens have numerous immunosuppressive effects such as decreasing proinflammatory cytokine release (e.g., IFNγ and TNF) or increasing anti-inflammatory cytokine release (e.g., IL-4 and IL-10), reducing T helper 1 (Th1) and T helper 17 (Th17) cell differentiation, inducing Treg differentiation and regulating B-cell development.(16-18) Paradoxically, these immunosuppressive effects of testosterone might be beneficial to overcome the heightened inflammatory environment that predisposes to severe COVID-19. Recent research has revealed that males with COVID-19 have lower testosterone levels.(19) Another study found a negative association between total testosterone levels and biochemical markers of COVID-19 severity.(20) Lower testosterone concentrations were associated with higher concentrations of IL-6, CRP, IL-1 receptor antagonist, hepatocyte growth factor, and IFNγ-inducible protein 10.(21) Therefore additional research efforts need to be made to investigate the complex relationships furtherly.”

Reference:

1. Fan X, Yin C, Wang J, Yang M, Ma H, Jin G, et al. Pre-diagnostic circulating concentrations of insulin-like growth factor-1 and risk of COVID-19 mortality: results from UK Biobank. Eur J Epidemiol. 2021;36(3):311-8.

2. Davey Smith G, Hemani G. Mendelian randomization: genetic anchors for causal inference in epidemiological studies. Hum Mol Genet. 2014;23(R1):R89-98.

3. Smith GD, Ebrahim S. 'Mendelian randomization': can genetic epidemiology contribute to understanding environmental determinants of disease? Int J Epidemiol. 2003;32(1):1-22.

4. Smith GD, Lawlor DA, Harbord R, Timpson N, Day I, Ebrahim S. Clustered environments and randomized genes: a fundamental distinction between conventional and genetic epidemiology. PLoS Med. 2007;4(12):e352.

5. Freuer D, Linseisen J, Meisinger C. Impact of body composition on COVID-19 susceptibility and severity: A two-sample multivariable Mendelian randomization study. Metabolism. 2021;118:154732.

6. Yavorska OO, Burgess S. MendelianRandomization: an R package for performing Mendelian randomization analyses using summarized data. Int J Epidemiol. 2017;46(6):1734-9.

7. Hemani G, Zheng J, Elsworth B, Wade KH, Haberland V, Baird D, et al. The MR-Base platform supports systematic causal inference across the human phenome. *eLife*. 2018;7.

8. Liu L, Fan X, Guan Q, Yu C. Bioavailable testosterone level is associated with COVID-19 severity in female: A sex-stratified Mendelian randomization study. J Infect. 2022;85(2):174-211.

9. Peckham H, de Gruijter NM, Raine C, Radziszewska A, Ciurtin C, Wedderburn LR, et al. Male sex identified by global COVID-19 meta-analysis as a risk factor for death and ITU admission. Nat Commun. 2020;11(1):6317.

10. Ho JQ, Sepand MR, Bigdelou B, Shekarian T, Esfandyarpour R, Chauhan P, et al. The immune response to COVID-19: Does sex matter? Immunology. 2022;166(4):429-43.

11. Lee KM, Heberer K, Gao A, Becker DJ, Loeb S, Makarov DV, et al. A Population-Level Analysis of the Protective Effects of Androgen Deprivation Therapy Against COVID-19 Disease Incidence and Severity. Front Med (Lausanne). 2022;9:774773.

12. Klein EA, Li J, Milinovich A, Schold JD, Sharifi N, Kattan MW, et al. Androgen Deprivation Therapy in Men with Prostate Cancer Does Not Affect Risk of Infection with SARS-CoV-2. J Urol. 2021;205(2):441-3.

13. Leach DA, Mohr A, Giotis ES, Cil E, Isac AM, Yates LL, et al. The antiandrogen enzalutamide downregulates TMPRSS2 and reduces cellular entry of SARS-CoV-2 in human lung cells. Nat Commun. 2021;12(1):4068.

14. Deng Q, Rasool RU, Russell RM, Natesan R, Asangani IA. Targeting androgen regulation of TMPRSS2 and ACE2 as a therapeutic strategy to combat COVID-19. iScience. 2021;24(3):102254.

15. Qiao Y, Wang XM, Mannan R, Pitchiaya S, Zhang Y, Wotring JW, et al. Targeting transcriptional regulation of SARS-CoV-2 entry factors ACE2 and TMPRSS2. Proc Natl Acad Sci U S A. 2020;118(1).

16. Olsen NJ, Kovacs WJ. Evidence that androgens modulate human thymic T cell output. J Investig Med. 2011;59(1):32-5.

17. Henze L, Schwinge D, Schramm C. The Effects of Androgens on T Cells: Clues to Female Predominance in Autoimmune Liver Diseases? Front Immunol. 2020;11:1567.

18. Trigunaite A, Dimo J, Jørgensen TN. Suppressive effects of androgens on the immune system. Cell Immunol. 2015;294(2):87-94.

19. Ma L, Xie W, Li D, Shi L, Ye G, Mao Y, et al. Evaluation of sex-related hormones and semen characteristics in reproductive-aged male COVID-19 patients. J Med Virol. 2021;93(1):456-62.

20. Rastrelli G, Di Stasi V, Inglese F, Beccaria M, Garuti M, Di Costanzo D, et al. Low testosterone levels predict clinical adverse outcomes in SARS-CoV-2 pneumonia patients. Andrology. 2021;9(1):88-98.

21. Dhindsa S, Zhang N, McPhaul MJ, Wu Z, Ghoshal AK, Erlich EC, et al. Association of Circulating Sex Hormones With Inflammation and Disease Severity in Patients With COVID-19. JAMA Netw Open. 2021;4(5):e2111398.

[Editors’ note: further revisions were suggested prior to acceptance, as described below.]

The manuscript has been improved but there are some remaining issues that need to be addressed, as outlined below:– The study hinges solely on MR analysis of selected SNPs used as proxies of testosterone, estradiol, SHBG, and IGF-1 levels. None of the markers were directly measured, and no information on their levels in the context of infections or COVID-19 was given. The authors argue at length that measuring the markers in the specific cohort is of limited value because their genetic causality modelling approach supersedes the need for such measurements. While MR models have indeed emerged as interesting tools to infer causality, it is also important to provide a potential mechanism linking genetic variants to the phenotype. Without direct measurements of the said markers minimize the potential to provide readers with a possible mechanism bridging genetic variant and phenotype.

Many thanks for your constructive suggestion. We agree with the reviewer that measurements of these biomarkers in study populations to verify the association are important. There are several epidemiological studies in which they measured IGF-1 levels in their study populations and followed up the clinical outcomes of COVID-19, and observed a significant association between low IGF-1 levels and severe COVID-19 outcomes. For instance, in the paper by Xikang Fan et al. (1), they demonstrated an inverse association between pre-diagnostic circulating levels of IGF-1 and COVID-19 mortality risk. Another observational study in Greece reported lower IGF-1 levels in critically ill COVID-19 patients compared to their counterparts with less severe disease or without COVID-19.(2)

Findings from these observational studies are the basis for the generation of hypothesis to be tested in this MR study. The adoption of MR design in this study aimed to provide different tiers of evidence to validate the observed association from previous studies. Considering the creativity and novelty of study findings, we highlighted the relevant findings from previous observational studies in the Discussion section, instead of repeating their work to explore the association between direct measurement of the marker levels and COVID-19 outcomes. We acknowledge that future studies, in particular laboratorial experiments, are warranted to investigate the possible mechanisms for the involvement of IGF-1 in the recovery of COVID-19. In order to provide some clues on the underlying mechanisms, we reviewed the relevant literature and tried to proposed some hypotheses to bridging up these findings.

Discussion section, page 9, line 209-214: “Our MR findings were consistent with the multiple epidemiological studies that reported an association between measured IGF-1 levels and COVID-19 illness. There is one observational study that demonstrated an inverse association between pre-diagnostic circulating levels of IGF-1 and COVID-19 mortality risk among COVID-19 patients in UK biobank. (26) Another observational study in Greece reported lower IGF-1 levels in critically ill COVID-19 patients compared to their counterparts with less severe disease or without COVID-19.(27) ”

Discussion section, page 9, line 220-239: “IGF-1 has been found to be pro-survival/anti-aging, anti-inflammatory, and antioxidant with neuro- and hepatoprotective properties. A study by the Narasaraju group demonstrated that IGF-1 plays an important role in the repair of lung tissue by regulating the proliferation and differentiation of alveolar epithelial cells (AECs).(29) Airway inflammation can be mitigated when apoptotic cells are engulfed by pulmonary epithelial cells.(30) IGF-1 has also been shown to up-regulate engulfment by professional phagocytes such as dendritic cells,(31) and inhibit IL-6 production from lipopolysaccharide (LPS)-induced AECs. (32) Both of these mechanisms are beneficial to the regression of local inflammation. Jakn et al. showed that IGF-1 binds to insulin-like growth factor-1 receptor (IGF-1R) on airway epithelial cells of non-professional phagocytic cells, which can promote the phagocytosis of microparticles by airway epithelial cells.(33) Transforming growth factor β1 (TGF-β1) derived from AECs activated alveolar macrophages (AMs) to secrete IGF-1 into the alveolar fluid in response to stimulation of the airway by inflammatory signals. This AM-derived IGF-1 attenuated the p38 mitogen-activated protein kinase (MAPK) inflammatory signal in AECs and promoted the phagocytosis of apoptotic cells by AECs. This two-way communication between AECs and AMs represents a well-tuned system for the regulation of the inflammatory response in alveoli.(34) Taken together, these studies provide biological evidence supporting that IGF-1 might be an important anti-inflammatory factor in the alveolar microenvironment and thus may contribute to improve COVID-19 outcomes. More studies are required to determine whether novel therapeutic strategy targeting on IGF-1 pathway might improve COVID-19 prognosis.”

Discussion section, page 11, line 291-293: “It should also be noted that the study findings are based on evidence from genetic data, additional large and prospective cohort studies with available IGF-1 data and information on COVID-19 susceptibility and clinical outcomes are needed to validate the findings.”

– The number of tests performed in the study is equivalent to the number of SNPs that were investigated, that is, 657. Hence, the multiple-testing correction must take into account 657 tests. The authors are wrong in adjusting their probabilities just for the 4 markers, which is more attuned to p-value hacking. Therefore, my initial comments on the lack of significant findings after considering multiple-test corrections remain a concern. Hence, the conclusions pertaining to IGF-1 variants do not reflect on the author's findings. Based on this result, I suggest being more objective by stating that none of the genetic variants can be deemed as a causal variant.

Many thanks for this comment and apologies for not addressing the previous comment properly. This Mendelian randomization study was designed to use genetic variants as instrumental variables to proxy the circulating level of 4 biomarkers, but not intends to examine the association between individual genetic variant and COVID-19 outcome. Thus, when applying the Bonferroni correction for multiple testing, the number of tests should be the number of biomarkers examined in the study, instead of the number of genetic variants used as IVs, therefore the adjusted P value should be < 0.05/4 = 0.013 (four biomarkers). The p-value for the association between IGF-1 and COVID-19 hospitalization was 0.018, which didn’t reach the significance threshold when adopting the Bonferroni correction. Given that there is also prior evidence for IGF-1 and COVID-19, this could be regarded as possibly a stand-alone prior hypothesis, in which case, P-value < 0.05 is acceptable for the validation of a prior hypothesis. However, to address this comment carefully, we have now revised the manuscript to describe the association as nominally significant and state this association did not survive the Bonferroni correction, and the strength of the conclusion has also been toned down. Revisions are made as below.

Abstract, page 2, line 49-53: “Conclusions: Our study indicated that genetically predicted high IGF-1 levels were associated with decrease the risk of COVID-19 susceptibility and hospitalization, but these associations did not survive the Bonferroni correction of multiple testing. Further studies are needed to validate the findings and explore whether IGF-1 could be a potential intervention target to reduce COVID-19 risk.”

Results, page 7, line 169-171: “Associations of IGF-1 levels with COVID-19 susceptibility and hospitalization were not statistically significant after Bonferroni correction, albeit showing a nominal significance at P<0.05.”

Discussion section, page 11, line 294-298: “In conclusion, our study indicated that genetically predicted high IGF-1 levels were associated with decrease the risk of COVID-19 susceptibility and hospitalization, but these associations did not survive the Bonferroni correction of multiple testing. Further studies are needed to validate the findings and explore whether IGF-1 could be a potential intervention target to reduce COVID-19 risk.”

References:

1. Fan X, Yin C, Wang J, Yang M, Ma H, Jin G, et al. Pre-diagnostic circulating concentrations of insulin-like growth factor-1 and risk of COVID-19 mortality: results from UK Biobank. Eur J Epidemiol. 2021;36(3):311-8.

2. Ilias I, Diamantopoulos A, Botoula E, Athanasiou N, Zacharis A, Tsipilis S, et al. Covid-19 and Growth Hormone/Insulin-Like Growth Factor 1: Study in Critically and Non-Critically Ill Patients. Front Endocrinol (Lausanne). 2021;12:644055.